# Determining optimal timing of birth for women with chronic or gestational hypertension at term: The WILL (When to Induce Labour to Limit risk in pregnancy hypertension) randomised trial

Laura A. Magee[1,2]*, Katie Kirkham[3], Sue Tohill[4], Eleni Gkini[3], Catherine A. Moakes[3], Jon Dorling[5], Marcus Green[6], Jennifer A. Hutcheon[7], Mishal Javed[3], Jesse Kigozi[3], Ben W. M. Mol[8], Joel Singer[9], Pollyanna Hardy[10], Clive Stubbs[3], James G. Thornton[11], Peter von Dadelszen[1,2], the WILL Trial Study Group[¶]

1 Department of Women and Children's Health, School of Life Course and Population Sciences, Faculty of Life Sciences and Medicine, King's College London, London, United Kingdom, 2 Institute of Women and Children's Health, King's College London, London, United Kingdom, 3 Birmingham Clinical Trials Unit, University of Birmingham, Birmingham, United Kingdom, 4 Maternity Services, Guy's and St Thomas' NHS Foundation Trust, London, United Kingdom, 5 Department of Paediatrics, University of Southampton, Southampton, United Kingdom, 6 Action on Pre-Eclampsia (APEC), Evesham, United Kingdom, 7 Department of Obstetrics and Gynaecology, University of British Columbia, Vancouver, Canada, 8 Department of Obstetrics and Gynaecology, Monash University, Clayton, Victoria, Australia, 9 School of Population and Public Health and Centre for Health Evaluation and Outcome Sciences, University of British Columbia, Vancouver, Canada, 10 National Perinatal Epidemiology Unit Clinical Trials Unit, University of Oxford, Oxford, United Kingdom, 11 Department of Obstetrics and Gynaecology, University of Nottingham, Nottingham, United Kingdom

¶ Membership of WILL Trial Study Group is provided in Table A in S3 Appendix.
* laura.a.magee@kcl.ac.uk

## Abstract

### Background

Chronic or gestational hypertension complicates approximately 7% of pregnancies, half of which reach 37 weeks' gestation. Early term birth (at 37 to 38 weeks) may reduce maternal complications, cesareans, stillbirths, and costs but may increase neonatal morbidity. In the WILL Trial (When to Induce Labour to Limit risk in pregnancy hypertension), we aimed to establish optimal timing of birth for women with chronic or gestational hypertension who reach term and remain well.

### Methods and findings

This 50-centre, open-label, randomised trial in the United Kingdom included an economic analysis. WILL randomised women with chronic or gestational hypertension at 36 to 37 weeks and a singleton fetus, and who provided documented informed consent to "Planned early term birth at 38$^{+0-3}$ weeks" (intervention) or "usual care at term" (control). The coprimary outcomes were "poor maternal outcome" (composite of severe hypertension, maternal death, or maternal morbidity; superiority hypothesis) and "neonatal care unit admission for

**Data Availability Statement:** Requests for data should be directed to BCTUdatashare@contacts. bham.ac.uk. Requests will be assessed for scientific rigour before being granted. Data will be anonymised and securely transferred. A data-sharing agreement might be required.

**Funding:** The trial was funded by the National Institute of Health Research (NIHR) Health Technology Assessment Programme (project number 16/167/123 to LAM, with co-applicants JD, JH, BWM, JS, PH, JGT, PvD, and from the WILL Study Group, PB, LC, MF, TR, and JS [PPIE]). The funder was not involved in study design, data collection or analysis, the decision to publish, or preparation of the manuscript.

**Competing interests:** The authors have declared that no competing interests exist.

**Abbreviations:** aRR, adjusted risk ratio; aRD, adjusted risk difference; BP, blood pressure; CHAP, Chronic Hypertension And Pregnancy; CHIPS, Control of Hypertension In Pregnancy Study; CI, confidence interval; NHS, National Health Service; NNTB, number-needed-to-treat-for-benefit; PPH, postpartum haemorrhage; TSC, Trial Steering Committee; WILL, When to Induce Labour to Limit risk in pregnancy hypertension.

≥4 hours" (noninferiority hypothesis). The key secondary was cesarean. Follow-up was to 6 weeks postpartum. The planned sample size was 540/group. Analysis was by intention-to-treat. A total of 403 participants (37.3% of target) were randomised to the intervention ($n$ = 201) or control group ($n$ = 202), from 3 June 2019 to 19 December 2022, when the funder stopped the trial for delayed recruitment. In the intervention (versus control) group, losses to follow-up were 18/201 (9%) versus 15/202 (7%). In each group, maternal age was about 30 years, about one-fifth of women were from ethnic minorities, over half had obesity, approximately half had chronic hypertension, and most were on antihypertensives with normal blood pressure. In the intervention (versus control) group, birth was a median of 0.9 weeks earlier (38.4 [38.3 to 38.6] versus 39.3 [38.7 to 39.9] weeks). There was no evidence of a difference in "poor maternal outcome" (27/201 [13%] versus 24/202 [12%], respectively; adjusted risk ratio [aRR] 1.16, 95% confidence interval [CI] 0.72 to 1.87). For "neonatal care unit admission for ≥4 hours," the intervention was considered noninferior to the control as the adjusted risk difference (aRD) 95% CI upper bound did not cross the 8% prespecified noninferiority margin (14/201 [7%] versus 14/202 [7%], respectively; aRD 0.003, 95% CI −0.05 to +0.06), although event rates were lower-than-estimated. The intervention (versus control) was associated with no difference in cesarean (58/201 [29%] versus 72/202 [36%], respectively; aRR 0.81, 95% CI 0.61 to 1.08). There were no serious adverse events. Limitations include our smaller-than-planned sample size, and lower-than-anticipated event rates, so the findings may not be generalisable to where hypertension is not treated with antihypertensive therapy.

## Conclusions

In this study, we observed that most women with chronic or gestational hypertension required labour induction, and planned birth at $38^{+0-3}$ weeks (versus usual care) resulted in birth an average of 6 days earlier, and no differences in poor maternal outcome or neonatal morbidity. Our findings provide reassurance about planned birth at $38^{+0-3}$ weeks as a clinical option for these women.

## Trial registration

isrctn.com ISRCTN77258279.

## Author summary

### Why was this study done?

- Approximately 7 in 100 women have high blood pressure that developed before pregnancy (chronic hypertension) or develops during pregnancy (gestational hypertension).

- Chronic or gestational hypertension may lead to more serious problems for women and babies, which is particularly common near the woman's due date.

- It may be possible to avoid such problems by timing birth a little earlier than the due date, as long as this does not create other problems for babies.

## What did the researchers do and find?

- In 50 hospitals in the United Kingdom, 403 (of a planned 1,080) women were recruited who had chronic or gestational hypertension, were otherwise well, and were nearing their due date.

- Women had an equal chance of being offered "Planned early term birth at $38^{+0-3}$ weeks" (intervention, 201 women) or "usual care at term" (control, 202 women) until December 2022 when the funder stopped the trial for delayed recruitment.

- Women in the intervention (compared with the control) group, gave birth 0.9 weeks earlier (at 38.4 weeks), and there was no evidence of a difference in "poor maternal outcome" (13% versus 12%, respectively) or "neonatal care unit admission for $\geq 4$ hours" (7% versus 7%, respectively). The intervention (versus control) was not associated with a difference in cesarean birth (29% versus 36%, respectively).

## What do these findings mean?

- Planned early term birth at $38^{+0-3}$ weeks may be the best option for women with chronic or gestational hypertension who are nearing their due date and remain well.

- While our findings are limited by a smaller-than-planned sample size, we were able to rule out differences in "poor maternal outcome" and "neonatal care unit admission for $\geq 4$ hours" that we specified as important before the trial.

- While it is possible that planned early term birth may reduce the occurrence of cesarean birth, further data will be required to confirm this.

## Introduction

Chronic or gestational hypertension complicates approximately 7% of pregnancies [1], half of which will reach 37 weeks' gestation [2]. There are no high-quality data on which to base timing of birth for this high-risk population.

Observational data suggest that early term birth (at 37 to 38 weeks) may reduce maternal complications (e.g., preeclampsia), cesareans, stillbirths [3–6], and costs, related primarily to a shorter duration of maternal-fetal surveillance [7]; however, early term birth may increase neonatal morbidity [8]. An individual patient data meta-analysis of randomised trials comparing early birth with expectant management in women with a hypertensive disorder of pregnancy also suggested earlier birth may benefit women without harming babies, including in subgroup analyses limited to participants with chronic and gestational hypertension [9]; however, these subgroup analyses included pregnancies randomised at preterm gestations, when the balance of harms and benefits associated with earlier birth are likely different. Also, the number of women enrolled in prior trials was small (e.g., 134 women with chronic hypertension), and the research was conducted in settings with differences in antenatal care, such as less frequent use of antihypertensive medication [10].

Timing of birth recommendations vary, demonstrating clinical equipoise. United Kingdom (UK) guidance advises timing of birth "be agreed between the woman and the senior obstetrician" [11]. International guidance states timed birth may be offered from $37^{+0}$ weeks (37

weeks and 0 days) for women with gestational hypertension and $38^{+0}$ weeks for those with chronic hypertension (weak recommendations) [1].

The WILL Trial (When to Induce Labour to Limit risk in pregnancy hypertension) aimed to establish optimal timing of birth for women with chronic or gestational hypertension who reach term gestational age and remain well.

## Methods

### Study design and participants

WILL was a 2-arm, parallel-group, open-label, multicentre, randomised trial in the UK (International Standard Randomised Controlled Trial Number, ISRCTN77258279; https://www.isrctn.com/ISRCTN77258279). The trial was approved by the National Health Service (NHS) Health Research Authority London Fulham Research Ethics Committee (reference 18/LO/2033). A 9-month internal pilot (3 June 2019 to 20 March 2020) tested trial processes in 20 centres; the Trial Steering Committee (TSC) and Data Monitoring Committee recommended the trial continue to the main phase. The protocol has been published [12] (**S1 Protocol**). This study is reported as per the CONsolidated Standards Of Reporting Trials (CONSORT) guideline and Guidelines for Reporting Trial Protocols and Completed Trials Modified Due to the COVID-19 Pandemic and Other Extenuating Circumstances (CONSERVE-CONSORT) (**S1 Consort Checklist**).

Participants were recruited from consultant-led UK maternity units. Women were eligible if they were ≥16 years of age, had chronic or gestational hypertension, and a live singleton fetus at $36^{+0}$ to $37^{+6}$ weeks. Hypertension was a systolic blood pressure (BP) ≥140 mm Hg or diastolic BP ≥90 mm Hg. Chronic hypertension was diagnosed before pregnancy or before 20 weeks, and gestational hypertension from 20 weeks [1]. Women were excluded if they had a contraindication to either trial group (e.g., preeclampsia), severe hypertension (systolic BP ≥160 mm Hg or diastolic BP ≥110 mm Hg) until resolved, a major fetal anomaly anticipated to require neonatal unit admission, or had consented to participate in another timed birth trial.

### Randomisation and masking

Women who provided documented informed consent were randomly assigned (1:1 ratio) to "planned early term birth at $38^{+0-3}$ weeks" (intervention) or "usual care at term" (control). Randomisation was by a central computerised service at the Birmingham Clinical Trials Unit and minimised for site, hypertension type, and prior cesarean. A "random element" was included in the minimisation algorithm, so that each woman had a probability of 20%, of being randomised to the opposite intervention that they would have otherwise received.

### Procedures

In the intervention group, birth could be initiated by labour induction or elective cesarean, by local protocol. In the control group, care was based on national guidance and local policy (as below) [11]. On 11 August 2022 (after randomisation of 348 women), the control group was changed from "expectant care until at least $40^{+0}$ weeks" to "usual care at term," to reflect practice change related to timed birth in other populations [13,14], the COVID-19 pandemic [15], and draft national labour induction guidance suggesting timed birth at 39 weeks may be appropriate for women at increased risk of term complications [16].

Adherence (defined in a binary context only in the intervention group) was timed birth initiation consistent with the allocated group, or earlier due to spontaneous onset of labour or birth for clinical need.

Both groups otherwise received standard maternity care [11], which included at least 1 antenatal visit at 38 weeks and another for nulliparous women at 40 weeks. Target BP was ≤135/85 mm Hg.

## Outcomes

Outcome data were abstracted from clinical notes. After birth, women were followed to 6 weeks postpartum.

The maternal coprimary outcome was a composite of poor maternal outcome until primary hospital discharge home or 28 days after birth (whichever was earlier), defined as severe hypertension, maternal death, or maternal morbidity, modelled on Delphi consensus [17] and the core outcome set in pregnancy hypertension [18] (for details, see **Table B** in S3 Appendix). This outcome was adjudicated by the local site principal investigator (or delegate), masked to allocated group and uninvolved in the woman's care, and based on review of primary case notes.

The neonatal coprimary outcome was neonatal care unit admission for ≥4 hours (resulting in separation of mother and baby), until primary hospital discharge home or 28 days after birth (whichever was earlier) [19].

The key secondary outcome was cesarean.

Other secondary outcomes included potential cointerventions, other pregnancy outcomes, maternal satisfaction, and healthcare resource use; for definitions, see **Table C** in S3 Appendix [12]. We included core outcomes in hypertensive pregnancy [18], except neonatal seizures. Adverse events were captured via predefined outcome measures in this high-risk population. A serious adverse event was one that resulted in death, was life-threatening, required hospitalisation or prolongation of existing hospitalisation, resulted in persistent or significant disability or incapacity, may have jeopardised the pregnancy, or may have required intervention to prevent one of the other outcomes listed above. Responses to the postpartum questionnaires will be reported elsewhere.

## Statistical analysis

We estimated that a sample size of 1,080 women (540/group) was required to detect a relative risk of 0.68, corresponding to an absolute risk reduction of 8% in poor maternal outcome, from a control group rate of 25% [20], 90% power, and two-sided alpha = 0.05 (superiority hypothesis). For the neonatal coprimary outcome, we estimated 540/group would have 88% power to detect a noninferiority margin of 8%, assuming a control group rate of 23% (i.e., the upper bound of the 95% confidence interval (CI) is <8%), two-sided alpha = 0.05, and 90% power to detect a 10% decrease in cesarean [10], assuming a control group rate of 45%. No adjustment was made for loss to follow-up or dropouts.

A statistical analysis plan (**S1 Appendix**) was developed before analyses which were intention-to-treat. Coprimary outcomes were analysed using mixed effects logistic regression, adjusted for hypertension type and prior cesarean as fixed effects (when convergence was possible), and recruiting centre as a random effect. Adjusted risk ratios (aRR) and adjusted risk differences (aRD) with 95% CIs were calculated by marginal standardisation for covariate adjustment [21]. For the neonatal coprimary outcome, noninferiority was based on the upper limit of the 95% CI in relation to our noninferiority margin of 8%. Binary secondary outcomes were analysed as per primary outcomes. Continuous outcomes were analysed using mixed effects linear regression to generate adjusted mean differences and 95% CIs.

Preplanned subgroup analyses were limited to coprimary outcomes and undertaken on variables used in the minimisation algorithm, except for recruiting centre; and ethnicity, body

mass index, prior severe hypertension (index pregnancy), or any of the following at randomisation: antihypertensive therapy, gestational diabetes mellitus, or smoking.

Sensitivity analyses limited to coprimary and key secondary outcomes were to assess the impact of missing data; further adjust for baseline characteristics; exclude women and babies if birth in the intervention arm was before $38^{+0-3}$ weeks, and in the control group before $39^{+0}$ weeks; assess heterogeneity of treatment effect due to the change to usual care (control arm); assess the impact on the neonatal coprimary outcome of stillbirths or neonatal deaths before neonatal unit admission. Unadjusted differences in medians (and corresponding 95% CI) were performed using bootstrapping methods (repetitions = 1,000, seed = 123,456). Complier Average Causal Effect analyses were not performed due to analytical difficulties in applying the standardisation approach.

The primary economic analysis was a cost-consequence analysis from a NHS perspective, comparing intervention and control management strategies. All resource use was valued with unit cost data (2020 to 2021 prices) obtained from NHS Reference Costs (**Table D** in S3 Appendix) [22]. Overall mean costs and their variance were calculated for outpatient visits, hospital admissions, tests of maternal-fetal well-being, maternity care, and neonatal care for both groups. Mean differences in costs were calculated using regression analysis, with bootstrapped bias-corrected 95% CIs (1,000 samples) (Health Economics Analysis Plan, **S2 Appendix**).

## Results

Among 50 participating sites with median 4,976 births annually (interquartile range: 3,400 to 5,800), 46 sites consented at least 1 woman from 3 June 2019 to 19 December 2022, with a pause after the internal pilot (20 March 2020 to 6 July 2020) due to the pandemic. During this, recruitment was delayed and the funder directed recruitment to stop, without knowledge of the results, as part of "post-pandemic reset."

Of 2,822 women screened, 1,030 were eligible, of whom 432 (42%) consented to participate (**Fig 1**); details of nonparticipation are in **Table D** in S3 Appendix. A total of 403 women were randomised, 201 to the intervention and 202 to the control group. There were 2 protocol deviations: inclusion in the control group of one woman with planned timed birth, and another who was randomised in error on the training randomisation website; both were analysed in their allocated group. Follow-up was complete for the coprimary outcomes, but 18/201 (9.0%) in the intervention and 15/202 (7.4%) in the control arms were lost to follow-up after hospital discharge, by 6 weeks postpartum.

Both groups were similar at trial entry (**Table 1**). On average, women were just over 30 years of age, with slightly more than one-fifth from ethnic minority groups and over half with BMI $\geq$30kg/m$^2$. Approximately half of women had chronic hypertension. Among 209 parous women (52%), approximately one-sixth had a prior cesarean. The gestational age at randomisation was just over 37 weeks. Most women were on antihypertensive medication at enrolment —almost always 1 agent, usually labetalol. BP was <140/90 mm Hg for most participants (**Tables 1** and **F** in S3 Appendix).

### Adherence

Adherence to the intervention was high (**Table 2**); nonadherence was most often due to busy hospital induction or theatre schedules. Gestational age at initiation of birth and gestational age at birth were each a median difference of 0.9 (95% CI 0.7 to 1.0; $p < 0.001$) weeks earlier in the intervention (versus control) group. The interval from initiation of birth to actual birth was a median difference of 0.3 weeks (95% CI 0.26 to 0.31) in the intervention group, and 0.3

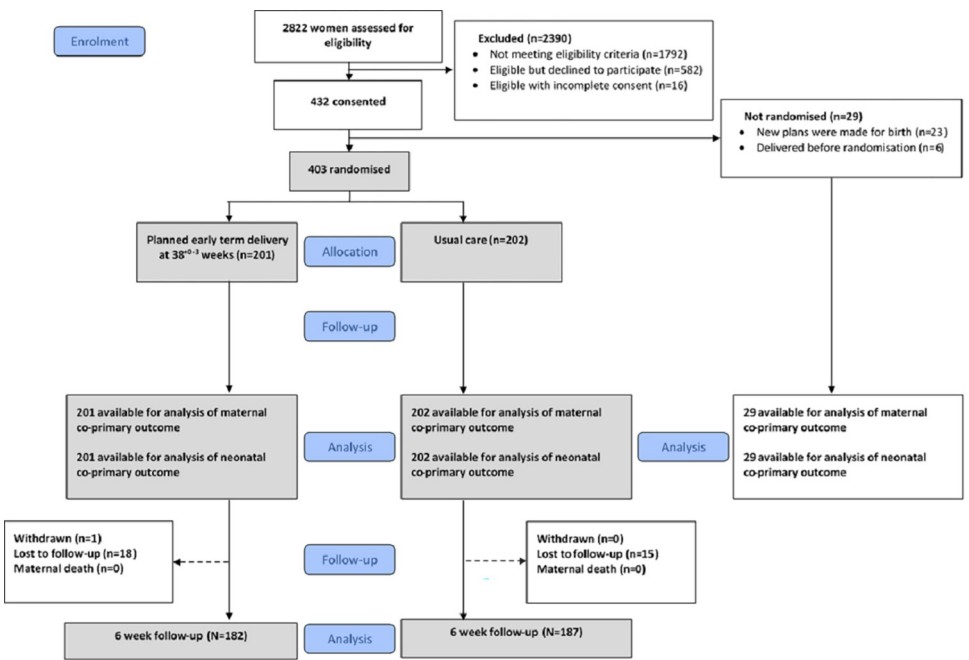

**Fig 1. Consort diagram of 1,030 eligible women from 46 sites, 598 (58.1%) did not consent to participation, 29 (2.8%) consented but were not randomised, and 403 (39.1%) consented and underwent randomisation.**

weeks (95% CI 0.14 to 0.43) in the control group. A minority of women in the control group went into spontaneous labour; most were induced. (Further details are in **Table G** in S3 Appendix).

## Outcomes

We found no evidence of a difference in the maternal coprimary ("poor maternal") outcome between intervention and control groups: 27/201 (13%) versus 24/202 (12%), respectively; aRR 1.16, 95% CI 0.72 to 1.87; $p = 0.538$ (**Table 2**). The 95% CI of the aRD (−0.05 to +0.09; $p = 0.539$) did not include the prespecified effect size of −0.08 (which corresponds to a "poor maternal" outcome event rate being 8% lower in the intervention versus control groups in absolute terms). There was evidence to suggest that receipt of transfusion (of any blood product), as a component of the composite outcome, occurred more often in the intervention (9/201 [4.5%]) versus control (2/202 [1.0%]) group, but the 95% CI reflected high levels of uncertainty due to low event rates (aRR 4.68, 95% CI 1.05 to 20.84; $p = 0.043$). All transfusions were postpartum, but there was no between-group difference evident in postpartum haemorrhage (PPH; see below).

For high-level neonatal care for ≥4 hours, the intervention group was considered noninferior to the control, as the upper bound of the aRD 95% CI did not cross the prespecified noninferiority margin of 8% (14/201 [7%] versus 14/202 [7%], aRD 0.003, 95% CI −0.05 to 0.06; $p = 0.912$); however, events rates were lower than estimated (**Table 2**). There were no stillbirths or neonatal deaths. High-level neonatal care was required most commonly for suspected/confirmed infection, respiratory disease, or "poor condition" at birth.

For coprimary outcomes, there was no evidence of heterogeneity of treatment effect by subgroup (**Table H** in S3 Appendix). Sensitivity analyses produced similar results (**Table I** in S3 Appendix).

**Table 1. Baseline characteristics (mean ± SD or N women (%) unless otherwise stated).**

| | Planned early term delivery at 38$^{+0-3}$ wks (N = 201) | Usual care at term (N = 202) |
|---|---|---|
| **Demographic and other baseline variables** | | |
| Maternal age at randomisation (years) | 31.5 ± 5.9 | 31.9 ± 5.7 |
| Mother's self-declared ethnicity | | |
| White | 157 (78.1) | 158 (78.2) |
| Black | 13 (6.5) | 17 (8.4) |
| Arab | 2 (1.0) | 1 (0.5) |
| South Asian | 16 (8.0) | 9 (4.5) |
| Other | 13 (6.5) | 16 (7.9) |
| Declined to give information | 0 | 1 (0.5) |
| Body mass index (kg/m$^2$) | | |
| <18.5 | 0 | 0 |
| 18.5–24.9 | 30 (14.9) | 35 (17.3) |
| 25.0–29.9 | 59 (29.4) | 48 (23.8) |
| ≥30 | 112 (55.7) | 119 (58.9) |
| **Hypertension type*** | | |
| Chronic | 96 (47.8) | 99 (49.0) |
| Gestational | 105 (52.2) | 103 (51.0) |
| Previous severe hypertension (sBP ≥160 mm Hg or dBP ≥110 mm Hg) during this pregnancy | 17 (8.5) | 25 (12.4) |
| **Prepregnancy medical and obstetric history** | | |
| Pregestational diabetes | 1 (0.5) | 4 (2.0) |
| Renal disease | 4 (2.0) | 5 (2.5) |
| Autoimmune disease (including APAS) | 9 (4.5) | 5 (2.5) |
| Nulliparous | 96 (47.8) | 98 (48.5) |
| In parous† women | (N = 105) | (N = 104) |
| Prior cesarean* | 15 (14.3) | 16 (15.4) |
| Prior gestational hypertension | 61 (58.1) | 56 (53.9) |
| Prior preeclampsia‡ | 29 (27.6) | 24 (23.1) |
| **This pregnancy** | | |
| Conceived by artificial reproductive technology‖ | 9 (4.5) | 6 (3.0) |
| Developed gestational diabetes in this pregnancy | 19 (9.5) | 18 (8.9) |
| Nicotine use after 20 wks of current pregnancy | 10 (5.0) | 13 (6.4) |
| Taking low-dose aspirin to prevent preeclampsia | 134 (66.7) | 134 (66.3) |
| **At trial enrolment** | | |
| GA at randomisation (wks) (median [IQR]) | 37.1 [37.0, 37.4] | 37.3 [37.0, 37.4] |
| **BP and antihypertensives at enrolment** | | |
| Taking antihypertensive medication at consent | 156 (77.6) | 165 (81.7) |
| Taking 1 agent | 146/156 (93.6) | 153/165 (92.7) |
| Agents taken¶ | | |
| Labetalol | 106/156 (68.0) | 128/165 (77.6) |
| Nifedipine | 32/156 (20.5) | 40/165 (24.2) |
| Methyldopa | 23/156 (14.7) | 5/165 (3.0) |
| Other** | 5/156 (3.2) | 5/165 (3.0) |
| Most recent sBP (mm Hg) before consent | 131.7 ± 11.2 | 132.9 ± 10.0 |
| Systolic <140 | 154 (76.6) | 151 (74.8) |
| Most recent dBP (mm Hg) before consent | 83.4 ± 8.3 | 83.1 ± 8.5 |
| Diastolic BP <90 | 155 (77.1) | 157 (77.7) |

(*Continued*)

**Table 1.** (Continued)

| | Planned early term delivery at 38^{+0−3} wks (N = 201) | Usual care at term (N = 202) |
|---|---|---|
| Device used to take BP | | |
| Automated device (any type) | 146 (72.6) | 143 (70.8) |
| Currently using home BP monitoring | 117 (58.2) | 110 (54.5) |

APAS, antiphospholipid antibody syndrome; BP, blood pressure; dBP, diastolic blood pressure; GA, gestational age; IQR, interquartile range (25th percentile, 75th percentile); sBP, systolic blood pressure; SD, standard deviation; wks, weeks.

*Minimisation variable, in addition to study site.

ⁱNumber of previous deliveries of fetus at $\geq 22^{+0}$ wks, $\geq 500$ g birthweight, or a crown-heel length $\geq 25$ cm.

‡Preeclampsia was defined as gestational hypertension with proteinuria or one/more relevant end-organ complications (https://www.nice.org.uk/guidance/ng133).

‖Defined as in vitro fertilisation with or without intracytoplasmic sperm injection, donor egg, or donor sperm.

¶Responses are not mutually exclusive.

**Other antihypertensive therapy in the intervention arm was amlodipine (N = 4) and felodopine, and in the control arm, amlodipine (N = 4) and hydralazine (N = 1).

There was no association between the intervention (versus control) for cesarean (58/201 [29%] versus 72/202 [36%], respectively; aRR 0.81, 95% CI 0.61 to 1.08; $p = 0.149$) (**Table 2**). However, the aRD and 95% CI (−0.07, 95% CI −0.16 to 0.02; $p = 0.146$) included the prespecified minimal clinically important difference of 10%. The trend towards a difference in cesarean was due to cesarean in labour (following spontaneous onset or induction, 18/201 [9%, intervention] versus 28/202 [14%, control]). The indication for cesarean in the intervention group was most often the study protocol (32/39 [82%]), and in the control group, based on maternal (30/72 [42%]) or fetal (30/72 [42%]) indications.

For the woman, there was no association between the intervention (versus control) and preeclampsia (56/201 [28%] versus 76/202 [38%], respectively; aRR 0.74, 95% CI 0.56 to 0.98; $p = 0.039$; **Table 3**), before and after birth; the absolute reduction translates into a number-needed-to-treat-for-benefit (NNTB) of 10. There was no association between the intervention (versus control) and PPH, sepsis, or intensive therapy unit admission. (For details, see **Tables 3** and **J** in S3 Appendix.)

For the baby, there was no association between the intervention (versus control) and respiratory problems (**Table 3**), regardless of definition, or other neonatal outcomes, including breastfeeding (**Tables 3** and **J** in S3 Appendix).

The intervention (versus control) was associated with less antihypertensive therapy use (**Table 3**); most women received 1 agent, usually labetalol. The intervention (versus control) was associated with less monitoring of well-being, with regard to preeclampsia blood or urine testing; outpatient visits by midwives or in the office/clinic, maternity assessment unit, or emergency department; or fetal cardiotocography or ultrasound. As such, over median [IQR] follow-up (to primary discharge home) of 10 days [8–12] in the intervention and 16 [11–20] in the control groups, the intervention (versus control) was associated with lower absolute rates of resource use and costs (mean ± SD): £6,659.57 ± 1,871.63 for the intervention and £7,067.37 ± 2,350.80 for the control groups (mean difference £−407.80, 95% CI −793.47 to +39.55; $p = 0.054$), with significantly lower costs for outpatient visits and tests of maternal-fetal well-being (**Table 4**; details, **Table K** in S3 Appendix). There were no serious adverse events.

## Discussion

For women with chronic or gestational hypertension who reach term and remain well, we found that planned early term birth at 38^{+0−3} weeks, versus usual care at term, resulted in birth

**Table 2. Adherence to the intervention and coprimary and key secondary outcomes, including stillbirth or neonatal death (median [IQR] or N (%)).**

| Outcomes | Planned early term delivery at 38$^{+0-3}$ wks (N = 201) | Usual care at term (N = 202) | Adjusted risk ratio† [95% CI]; p-value | Adjusted risk difference‡ [95% CI]; p-value |
|---|---|---|---|---|
| **Adherence** | | | | |
| Adherent* | 184 (91.5) | NA | NA | NA |
| Reasons for nonadherence: | (N = 17) | - | NA | NA |
| Busy hospital induction or theatre schedules | 11/17 (64.6) | - | NA | NA |
| Womens' preference | 2/17 (11.8) | - | NA | NA |
| Clinicians' preference | 1/17 (5.9) | - | NA | NA |
| Spontaneous birth at GA above 38$^{+3}$ wks | 2/17 (11.8) | - | NA | NA |
| Withdrawal from treatment | 1/17 (5.9) | - | NA | NA |
| GA at initiation of birth (induction or no labour) | 38.1 [38.0, 38.3] | 39.0 [38.6, 39.7] | NA | NA |
| GA at birth (all women) | 38.4 [38.3, 38.6] | 39.3 [38.7, 39.9] | NA | NA |
| Method of delivery initiation | | | | |
| Spontaneous onset of labour | 8 (4.0) | 45 (22.3) | NA | NA |
| No labour (elective cesarean) | 18 (8.9) | 18 (8.9) | NA | NA |
| Induced | 175 (87.1) | 139 (68.8) | NA | NA |
| **Maternal coprimary: "Poor maternal outcome"¶** | 27 (13.4%) | 24 (11.9%) | 1.16 [0.72 to 1.87]; 0.538 | 0.02 [−0.05 to 0.09]; 0.539 |
| Components of "poor maternal outcome" | | | | |
| sBP ≥160 mm Hg or dBP ≥110 mm Hg | 17 (8.5) | 19 (9.4) | 0.95 [0.53 to 1.72]; 0.869 | −0.005 [−0.06 to 0.05]; 0.869 |
| Pulmonary oedema | 1 (0.5) | 0 | Not estimable | Not estimable |
| SpO$_2$ <90% | 3 (1.5) | 0 | Not estimable | Not estimable |
| Acute kidney injury or dialysis | 2 (1.0) | 1 (0.5) | 2.01¥ [0.18 to 22.00]; 0.567 | 0.01¥ [−0.01 to 0.02]; 0.559 |
| Placental abruption | 1 (0.5) | 3 (1.5) | 0.34§ [0.04 to 3.20]; 0.342 | −0.01§ [−0.03 to 0.01]; 0.316 |
| Transfusion‖ | 9 (4.5) | 2 (1.0) | 4.68 [1.05 to 20.84]; 0.043 | 0.04 [0.001 to 0.08]; 0.045 |
| Vaginal birth (noninstrumental) | 4/9 | 2/2 | NA | NA |
| Vaginal birth (instrumental) | 3/9 | 0/2 | NA | NA |
| Cesarean before labour | 0/9 | 0/2 | NA | NA |
| Cesarean in labour | 2/9 | 0/2 | NA | NA |
| **Neonatal coprimary: Neonatal care unit admission for ≥4 hours** | 14 (7.0) | 14 (6.9) | 1.03§ [0.52 to 2.08]; 0.912 | 0.003§ [−0.05 to 0.06]; 0.912** |
| Stillbirth | 0 | 0 | Not estimable | Not estimable |
| Neonatal death | 0 | 0 | Not estimable | Not estimable |
| Indications for high-level neonatal care for ≥4 hours*** | (N = 14) | (N = 14) | | |
| Infection (suspected/confirmed) | 10/14 (71.4) | 8/14 (57.1) | NA | NA |
| Respiratory disease | 5/14 (28.6) | 10/14 (71.4) | NA | NA |
| Poor condition at birth | 2/14 (35.7) | 3/14 (21.4) | NA | NA |
| Hypoglycaemia | 2/14 (14.3) | 1/14 (7.1) | NA | NA |
| Other†† | 1/14 (7.1) | 1/14 (7.1) | NA | NA |
| **Key secondary outcome: Cesarean birth** | 58 (28.9) | 72 (35.6) | 0.81 [0.61 to 1.08]; 0.149 | −0.07 [−0.16 to 0.02]; 0.146 |
| Type of cesarean | | | | |
| No labour ("elective")‡‡ | 18 (9.0) | 18 (8.9) | NA | NA |
| Nonelective cesarean | 40 (19.9) | 54 (26.7) | NA | NA |
| Following spontaneous onset labour¥¥ | 1 (0.5) | 10 (5.0) | NA | NA |

(*Continued*)

**Table 2.** (Continued)

| Outcomes | Planned early term delivery at $38^{+0-3}$ wks ($N = 201$) | Usual care at term ($N = 202$) | Adjusted risk ratio† [95% CI]; p-value | Adjusted risk difference‡ [95% CI]; p-value |
|---|---|---|---|---|
| Following labour induction | | | | |
| No labour‡‡ | 21 (10.5) | 16 (7.9) | NA | NA |
| In labour¥¥ | 18 (9.0) | 28 (13.9) | NA | NA |
| Indications for cesarean | | | | |
| Dictated by study protocol§§ | 32/39 (82.1) | 8/34 (23.5) | NA | NA |
| Maternal | 12/58 (20.7) | 30/72 (41.7) | NA | NA |
| Fetal | 11/58 (19.0) | 30/72 (41.7) | NA | NA |
| Busy hospital induction/theatre schedules§§ | 4/39 (10.3) | 1/34 (2.9) | NA | NA |
| Woman's preference | 1/58 (1.7) | 4/72 (5.6) | NA | NA |
| Clinicians' preference | 0/58 (0) | 2/72 (2.8) | NA | NA |
| Other | 7/58 (12.1) | 19/72 (26.4) | NA | NA |
| Vaginal birth | | | | |
| Vaginal (noninstrumental) | 122 (60.7) | 107 (53.0) | NA | NA |
| Vaginal birth (instrumental) | 21 (10.4) | 23 (11.4) | NA | NA |

ARM, artificial rupture of membranes; BP, blood pressure; dBP, diastolic blood pressure; GA, gestational age; GDM, gestational diabetes; IQR, interquartile range as (25th percentile, 75th percentile); NA, not applicable; sBP, systolic blood pressure; SAP, statistical analysis plan; SpO₂, oxygen saturation; wks, weeks.

*Adherence was defined as timing of delivery initiation consistent with the allocated group or if earlier, delivery timing as a result of either spontaneous onset of labour or delivery for clinical need. This was defined as a binary variable only in the intervention group.

†Risk ratio was adjusted for minimisation variables (centre, hypertension type and prior cesarean) as categorical covariates, with centre included as a random effect. A value <1 favours planned early term delivery. The p-value was generated from the marginal standardisation model, which followed the mixed effects logistic regression.

‡Risk difference was adjusted for minimisation variables (centre, hypertension type and prior cesarean) as categorical covariates, with centre included as a random effect. A value <0 favours planned early term delivery. The p-value was generated from the marginal standardisation model, which followed the mixed effects logistic regression.

#GA at initiation of birth refers to the GA at the start of labour induction or elective cesarean. GA at birth refers to the date of delivery.

¶There were none of the following poor maternal outcomes: maternal death, Glasgow Coma Scale <13, stroke, transient ischaemic attack, eclampsia, blindness, uncontrolled hypertension, inotropic support, respiratory failure, myocardial ischaemia/infarction, hepatic dysfunction, hepatic haematoma/rupture, platelet count <50 × 10⁹/L.

¥For this model, hypertension type and prior cesarean were removed due to convergence issues.

§For this model, prior cesarean was removed due to convergence issues.

‖All transfusions were administered after birth, a median [IQR] of 0 [0, 0] vs. 0.5 [0, 1] days postpartum.

**This is the p-value for superiority from the adjusted analysis. Noninferiority has been achieved because the upper boundary of the 95% CI of the neonatal outcome is less than the 8% absolute difference margin.

***Indications for neonatal care unit admission for ≥4 hours was based on the electronic health record discharge summary for the neonate's first admission.

††Other indications for high-level neonatal care for ≥4 hours in the intervention group ($N = 1$) was: jaundice from ABO isoimmunisation; and in the control group ($N = 1$): difference between pre and post ductal SaO2.

‡‡"No labour ('elective') and 'Following labour induction/No labour" together constitute "Before labour" in the SAP.

¥¥"Following spontaneous onset of labour" and "Following labour induction/In labour" constitute "In labour" in the SAP.

§§These indications are relevant only for cesarean before labour.

an average of 6 days earlier, although almost 70% of women in the usual care at term group still required labour induction. Planned early term birth at $38^{+0-3}$ weeks resulted in lower-than-anticipated rates of adverse maternal and fetal/newborn coprimary outcomes, with no evidence of differences compared with usual care at term. The 95% CI around our comparative estimate for the maternal coprimary outcome excludes our target difference of a 32% relative risk reduction and an 8% absolute risk reduction. Similarly, the 95% CI around our

**Table 3. Other pregnancy outcomes and cointerventions (N (%) or median [IQR]).**

| Outcomes | Planned early term delivery at 38$^{+0-3}$ wks (N = 201) | Usual care at term (N = 202) | Adjusted risk ratio* [95% CI]; p-value | Adjusted risk difference‡ [95% CI]; p-value |
|---|---|---|---|---|
| **Other maternal outcomes** | | | | |
| Preeclampsia‡ | 56 (27.9) | 76 (37.6) | 0.74 [0.56 to 0.98]; 0.039 | −0.10 [−0.19 to −0.01]; 0.036 |
| Before birth | 40 (19.9) | 54 (26.7) | NA | NA |
| Gestational age (wks) | 38.0 [37.6, 38.2] | 38.3 [37.7, 39.1] | NA | NA |
| After birth | 16 (8.0) | 22 (10.9) | NA | NA |
| Elevated AST or ALT (>40 IU/L) | 7 (3.5) | 13 (6.4) | 0.54 [0.22 to 1.33]; 0.183 | −0.03 [−0.07 to 0.01]; 0.176 |
| Platelet count <100 × 10$^9$/L | 1 (0.5) | 1 (0.5) | 1.01‖ [0.06 to 15.97]; 0.997 | 0.00003‖ [−0.01 to 0.01]; 0.997 |
| Mode of birth | | | | |
| Vaginal (noninstrumental) | 122 (60.7) | 107 (53.0) | 0.84** [0.67 to 1.05]; 0.121 | −0.08** [−0.17 to 0.02]; 0.118 |
| Vaginal birth (instrumental) | 21 (10.4) | 23 (11.4) | | |
| Cesarean (no labour) | 39 (19.4) | 34 (16.8) | | |
| Cesarean (in labour) | 19 (9.5) | 38 (18.8) | | |
| Postpartum haemorrhage | 29 (14.4) | 33 (16.3) | 0.89 [0.57 to 1.40]; 0.623 | −0.02 [−0.09 to 0.05]; 0.623 |
| Sepsis | 2 (1.0) | 0 (0) | Not estimable | Not estimable |
| Intensive therapy unit admission | 3 (1.5) | 0 (0) | Not estimable | Not estimable |
| **Other neonatal outcomes** | | | | |
| Birthweight <10th centile | 8 (4.0) | 12 (5.9) | NA | NA |
| 5-min Apgar score <7 | 5/199 (2.5) | 5/201 (2.5) | NA | NA |
| Respiratory problems | | | | |
| As indication for high-level neonatal care for ≥4 hrs | 4 (2.0) | 7 (3.5) | 0.59‖ [0.18 to 1.95]; 0.390 | −0.02‖ [−0.05 to 0.02]; 0.384 |
| Requiring intervention‖‖‖ | 5 (2.5) | 10 (5.0) | 0.51‖ [0.18 to 1.46]; 0.208 | −0.02‖ [−0.06 to 0.01]; 0.196 |
| Oxygen given | 5 (2.5) | 10 (5.0) | NA | NA |
| Positive pressure ventilation | 1 (0.5) | 7 (3.5) | NA | NA |
| Defined clinically¶¶ | 6 (3.0) | 9 (4.5) | 0.67‖ [0.24 to 1.86]; 0.445 | −0.01‖ [−0.05 to 0.02]; 0.441 |
| Chest X-ray performed | 6 (3.0) | 7 (3.5) | 0.87‖ [0.30 to 2.54]; 0.799 | −0.004‖ [−0.04 to 0.03]; 0.799 |
| Abnormal X-ray, n (%)*** | 1/6 (16.7) | 1/7 (14.3) | NA | NA |
| Hypoxic-ischaemic encephalopathy | 0 (0) | 1 (0.5) | Not estimable | Not estimable |
| Sepsis requiring antibiotics for at least 5 days | 2 (1.0) | 5 (2.5) | 0.40‖ [0.08 to 2.04]; 0.271 | −0.01‖ (−0.04 to 0.01); 0.253 |
| Breastfeeding established | 125 (62.2) | 115 (56.9) | 1.09 [0.93 to 1.28]; 0.277 | 0.05 (−0.04 to 0.15); 0.276 |
| Exclusive breastfeeding | 90 (45.0) | 87 (43.1) | 1.05 [0.84 to 1.30]; 0.689 | 0.02 (−0.08 to 0.12); 0.689 |
| Antihypertensive therapy | | | | |
| Antepartum | 166 (82.6) | 182 (90.1) | 0.93 [0.87 to 0.99]; 0.029 | −0.07 [−0.12 to −0.01]; 0.025 |
| Taking 1 agent | 135 (81.3) | 143 (78.6) | NA | NA |
| Taking 2 or more agents | 31 (18.7) | 39 (21.4) | NA | NA |
| Agents taken | | | | |
| Labetalol | 124 (74.7) | 153 (84.1) | NA | NA |
| Methyldopa | 21 (12.7) | 7 (3.9) | NA | NA |

*(Continued)*

**Table 3.** (*Continued*)

| Outcomes | Planned early term delivery at 38$^{+0-3}$ wks (N = 201) | Usual care at term (N = 202) | Adjusted risk ratio* [95% CI]; p-value | Adjusted risk difference‡ [95% CI]; p-value |
|---|---|---|---|---|
| Nifedipine long-acting | 2 (1.2) | 5 (2.8) | NA | NA |
| Nifedipine modified-release | 44 (26.5) | 54 (29.7) | NA | NA |
| Other | 7 (4.2) | 5 (2.8) | NA | NA |
| Postpartum | 148 (73.6) | 174 (86.1) | 0.86 [0.79 to 0.95]; 0.002 | −0.12 [−0.19 to −0.05]; 0.002 |
| Taking 1 agent | 122 (82.4) | 145 (83.8) | NA | NA |
| Taking 2 or more agents | 26 (17.6) | 28 (16.2) | NA | NA |
| *Missing* | 0 | 1 | | |
| Agents taken | | | NA | NA |
| Labetalol | 102 (68.9) | 127 (73.0) | NA | NA |
| Methyldopa | 6 (4.1) | 6 (3.5) | NA | NA |
| Nifedipine long-acting | 3 (2.0) | 2 (1.2) | NA | NA |
| Nifedipine modified-release | 37 (25.0) | 46 (26.4) | NA | NA |
| Other | 28 (18.9) | 23 (13.2) | NA | NA |
| Both antepartum and postpartum | 168 (83.6) | 184 (91.1) | 0.93 [0.87 to 0.99]; 0.026 | −0.07 [−0.12 to −0.01]; 0.023 |
| Magnesium sulphate | 7 (3.5) | 3 (1.5) | 2.34 [0.61 to 8.90]; 0.214 | 0.02 [−0.01 to 0.05]; 0.199 |
| Use of home BP monitoring | 114 (57.0) | 115 (56.9) | 1.00 [0.84 to 1.19]; 0.983 | 0.001 [−0.09 to 0.09]; 0.983 |
| Bedrest at home | 2 (1.0) | 0 (0) | Not estimable | Not estimable |
| Preeclampsia blood/urine testing before delivery admission | 81 (40.5) | 125 (61.9) | 0.65 [0.53 to 0.79]; <0.001 | −0.22 [−0.31 to −0.12]; <0.001 |
| Outpatient visits (in office/clinic)¥¥ | 94 (47.0) | 132 (65.4) | 0.72 [0.60 to 0.85]; <0.001 | −0.18 [−0.28 to −0.09]; <0.001 |
| Outpatient visits (in woman's home)¥¥ | 29 (14.5) | 38 (18.8) | 0.75 [0.50 to 1.11]; 0.151 | −0.05 [−0.12 to 0.02]; 0.150 |
| Medical, day or maternity assessment unit visits | 98 (49.0) | 143 (70.8) | 0.69 [0.58 to 0.81]; <0.001 | −0.22 [−0.31 to −0.13]; <0.001 |
| Seen in acute care area for urgent/emergent visit other than in labour | 4 (2.0) | 10 (4.9) | 0.36 [0.12 to 1.10]; 0.073 | −0.03 [−0.06 to 0.005]; 0.095 |
| Admission days prior to admission for birth | 0 [0, 0] | 0 [0, 0] | 1.15¥ [0.42 to 3.12]; 0.786 | NA |
| None | 191 (95.5) | 189 (93.6) | NA | NA |
| One | 5 (2.5) | 9 (4.4) | NA | NA |
| Two | 2 (1.0) | 2 (1.0) | NA | NA |
| Three or more | 2 (1.0) | 2 (1.0) | NA | NA |
| *Missing* | 1 | 0 | NA | NA |
| Fetal cardiotocography | 93 (46.5) | 131 (64.9) | 0.70 [0.58 to 0.84]; <0.001 | −0.19 (−0.28 to −0.10); <0.001 |

(*Continued*)

**Table 3.** (Continued)

| Outcomes | Planned early term delivery at $38^{+0-3}$ wks (N = 201) | Usual care at term (N = 202) | Adjusted risk ratio* [95% CI]; p-value | Adjusted risk differenceɫ [95% CI]; p-value |
|---|---|---|---|---|
| Fetal ultrasound | 41 (20.5) | 85 (42.1) | 0.49 [0.36 to 0.67]; <0.001 | −0.22 (−0.30 to −0.13); <0.001 |

ALT, alanine aminotransferase; AST, aspartate aminotransferase; BW, birthweight; hrs, hours; IQR, interquartile range as (25th percentile, 75th percentile); IRR, incidence rate ratio; NA, not applicable; SD, standard deviation; wks, weeks.

*Risk ratio was adjusted for minimisation variables (centre, hypertension type and prior cesarean) as categorical covariates, with centre included as a random effect. A value <1 favours planned early term delivery. The p-value was generated from the marginal standardisation model, which followed the mixed effects logistic regression.

ɫRisk difference and mean difference were adjusted for minimisation variables (centre, hypertension type and prior cesarean) as categorical covariates, with centre included as a random effect. A value <0 favours planned early term delivery. The p-value was generated from the marginal standardisation model, which followed the mixed effects logistic regression.

‡There were none of the following preeclampsia criteria met: Glasgow Coma Scale <13, stroke, eclampsia, blindness, clonus, platelet count <$50 \times 10^9$/L, disseminated intravascular coagulation, haemolysis, abnormal umbilical artery Doppler, or stillbirth.

‖For this model, prior cesarean was removed due to convergence issues.

**Instrumental vaginal delivery or cesarean delivery vs. noninstrumental vaginal delivery.

‖‖Respiratory morbidity was defined as the need for supplemental oxygen and/or positive pressure ventilation beyond the initial resuscitation period.

¶¶Clinical respiratory problem was defined as: transient tachypnoea of newborn (0 in intervention vs. 7 in control groups), meconium aspiration syndrome (0 vs. 1, respectively), pneumonia (0 vs. 0, respectively), pneumothorax/pneumomediastinum (1 vs. 0, respectively), or other (6 vs. 3, respectively).

***The abnormal chest X-ray findings were pneumothorax/pneumomediastinum (1 in intervention) and right lung field more hazy than left (1 in control). In v3.0 of the protocol, to better define respiratory disease and match existing data collected, we added, "Chest X-ray, N performed, N abnormal and nature of abnormality (i.e., meconium aspiration syndrome, pneumonia, pneumothorax/pneumomediastinum, transient tachypnoea of newborn, or other [unspecified]).

¥Incidence rate ratio adjusted for minimisation variables (hypertension type and prior cesarean) as categorical covariates, using a negative binomial model as data were dispersed (95% CI of the dispersion parameter: .5.69 to 25.92). Centre was excluded from the model due to convergence issues. The natural logarithm of time in days from the date of randomisation to the date of admission for birth is added as an offset variable. A value<1 favours planned early term delivery.

¥¥Clarification that outpatient visits could be in the office/clinic or in the woman's home was made in v3.0 of the protocol (5 November 2020), although the data were collected like this throughout the trial.

comparative estimate for the neonatal coprimary outcome excludes our target noninferiority margin of 8% increase in risk.

Also, we found that planned early term birth (versus usual care at term) was associated with no increase (and potential reduction) in cesarean, with the 95% CI from the comparative estimate that included a 10% reduction in risk set as the minimally-clinically important difference

**Table 4. Cost analysis in British pounds.**

| Costs | Planned early term birth (N = 201) | | Usual care at term (N = 202) | | Difference in mean costs (intervention minus control groups) (95% CI) | P value* |
|---|---|---|---|---|---|---|
| | Mean | SD | Mean | SD | | |
| Outpatient visits | 302.92 | 425.87 | 538.23 | 418.71 | −235.32 (−309.45 to −154.13) | 0.000 |
| Inpatient admissions | 1,043.34 | 613.35 | 956.97 | 477.05 | 86.37 (−15.34 to +200.41) | 0.110 |
| Tests of maternal or fetal well-being | 156.44 | 170.61 | 259.29 | 188.39 | −102.84 (−136.65 to −67.78) | 0.000 |
| Obstetric care | 5,010.96 | 1,440.97 | 5,049.80 | 1,713.52 | −38.84 (−344.97 to +275.03) | 0.807 |
| Neonatal care | 145.91 | 549.65 | 263.07 | 1,018.34 | −117.16 (−281.76 to +38.42) | 0.152 |
| Total costs | 6,659.57 | 1,871.63 | 7,067.37 | 2,350.80 | −407.80 (−793.47 to +39.55) | 0.054 |

CI, confidence interval; SD, standard deviation.

*Costs were compared between groups by regression analysis, with bootstrapped bias-corrected 95% CIs.

a priori. The intervention (versus control) was associated with a reduction in preeclampsia (defined broadly) for the woman (NNTB = 10), with no evidence of increased health problems for the baby. Also, the intervention (versus control) was associated with lower healthcare utilisation (monitoring of maternal-fetal well-being, and obstetric outpatient visits) and associated costs, with the direction of costs overall favouring planned early term birth.

To the best of our knowledge, WILL is the largest randomised evaluation of timed birth for women with chronic or gestational hypertension at term. Most prior trials enrolled women with preeclampsia and have dominated meta-analyses of timed birth for women with pregnancy hypertension [9,23–25]. While 4 trials have included at least some women who would have met WILL eligibility criteria (at least 340 participants), only 1 trial excluded women with preeclampsia [26]. That trial was small ($N = 102$), not prospectively registered, and found no differences in a composite of maternal/neonatal mortality/morbidity or cesarean. There is one ongoing trial (250 women) in India of timed birth at 38 (versus 40) weeks for mild gestational hypertension (CTRI/2022/06/043028, recruitment anticipated to end in 2024).

Our findings are consistent with observational data suggesting $38^{+0}$ to $39^{+6}$ weeks is the optimal timing of birth for women with chronic or gestational hypertension at term [3,4]. Our observed trend towards a reduction in cesarean associated with planned early term birth is consistent with the HYPITAT trial [10] and trials of induction for other indications [27,28]. While we observed an increase in transfusion associated with planned early term birth (versus usual care at term), the 95% CI ranged from 0.1% to 8.0% increased risk, reflecting substantial uncertainty. There was no evidence of an increase in PPH, consistent with systematic reviews of labour induction for either any indication (versus expectant care) at term [29], or for pregnancy hypertension, including chronic or gestational hypertension [9].

The WILL trial demonstrated low rates of maternal and fetal/newborn morbidities at term for women with chronic or gestational hypertension. This may be due to WILL being undertaken in the current era of good BP control [11]. Most women in WILL were taking antihypertensive therapy at enrolment and had BP <140/90 mm Hg. Improved maternal outcomes are consistent with the reduction in severe hypertension and maternal end-organ complications that define preeclampsia (e.g., thrombocytopoenia), as seen with BP control in the Control of Hypertension In Pregnancy Study (CHIPS) and the Chronic Hypertension And Pregnancy (CHAP) trials [2,30]. A recent retrospective cohort study of timed birth in women with chronic hypertension controlled with antihypertensive therapy found similarly low adverse outcome rates [31].

Our healthcare utilisation and economic findings are similar to those of the HYPITAT trial, in which earlier birth was associated with a shorter duration of (and less overall) maternal-fetal surveillance, and lower associated costs [7].

A strength of the trial was its generalisability to real-world care of women with chronic or gestational hypertension, through inclusion of women with comorbidities, contemporary treatment (control) of hypertension with antihypertensive therapy, and comparison of planned early term birth with usual clinical practice. Of note, the trial was reviewed independently and found to exceed expectations for having a diverse study population [32].

Our major limitation is that we reached only 37% of our recruitment target before trial cessation by the funder. Nevertheless, to the best of our knowledge, WILL is the largest randomised evaluation of timed birth for this group of women who reach term gestational age and remain well. It is likely that no more than 122 women in the HYPITAT trial would have been eligible for WILL, given that (i) 65% of participants had gestational hypertension; (ii) they were recruited at the time that they developed that gestational hypertension; and (iii) a minority (188/756) of participants overall were recruited at $37^{+0-6}$ weeks, when they were randomised to labour induction within 24 hours or ongoing expectant care (whereas women in WILL were randomised to planned timed birth at $38^{+0-3}$ weeks versus usual care). We included

women with either chronic or gestational hypertension; while recruiting a population of mixed hypertensive type is common in pregnancy hypertension trials [2] and results for the coprimary outcomes were similar by hypertension type, such subgroup analyses were predictably underpowered. While the label of our control arm was changed to "usual care at term," this applied only to the final 14% of recruits, and throughout, the control group reflected current practice. The findings may not be generalisable to where hypertension is not treated with antihypertensive therapy, despite international recommendations. The event rates of the 2 coprimary outcomes were lower than anticipated; while the relative risk of 0.68 set a priori for the maternal coprimary outcome was excluded, and the clinically important changes specified in absolute risks were also achieved, those changes in absolute risk for the maternal (8% reduction) and neonatal (8% noninferiority margin) coprimary outcomes were unrealistically large given the lower-than-anticipated event rates. Finally, we did not collect information on the level of neonatal care required.

The WILL trial results indicate that for women with chronic or gestational hypertension whose BP is controlled, who have reached term, and remain well, most (78%) women managed expectantly will require iatrogenic birth for clinical need, prior to the onset of spontaneous labour. While the likelihood is low that planned early birth is harmful for the baby, such a management strategy may be beneficial for women. Planned early term birth is associated with a clinically important, lower risk of progression to preeclampsia, albeit potentially, associated with a smaller, increased risk of transfusion; this stands alone as an intervention that could reduce the risk of progression to preeclampsia at term in women with chronic or gestational hypertension, similar to development of de novo preeclampsia in the ARRIVE trial of timed birth at term for low-risk nulliparous women [13]. The potential reduction in cesarean may appeal to women, and the associated reduction in healthcare utilisation and some health system costs may prompt some units to recommend planned early term birth to these women. Thus, it appears on balance that planned early term birth at $38^{+0-3}$ weeks may be the preferred clinical option.

Future work should address whether planned early term birth in women with chronic or gestational hypertension reduces cesarean; an individual participant data meta-analysis is planned (CRD42024498376), as it would be difficult to justify mounting another randomised trial given our low adverse event rates affecting feasibility. Also, observational data have raised concerns that in the general population, across the whole range of gestational at birth, gestational age has a strong, dose-dependent relationship with special educational needs, including mild learning disabilities such as dyslexia and attention deficit hyperactivity disorder [33]. While data from nonrandomised comparisons of labour induction or expectant care at term have been reassuring with regard to neurodevelopmental outcomes [34,35], condition-specific data are needed. WILL participants were asked for consent for collection of routinely collected health data, including those measuring school performance.

Pending definitive data, the WILL trial findings provide reassurance about planned early term birth at $38^{+0-3}$ weeks as a clinical option for women with chronic or gestational hypertension who reach this gestational age undelivered.

## Patient and public involvement (PPIE)

The trial had 2 PPIE coapplicants (MG, JS), a PPIE representative on the TSC, and a bespoke PPIE group (Emma Jukes, Fatima Rami, Al Richards, Khilna Rupen, Debs Smith) that reviewed patient and public-facing material for trial promotion and recruitment.

## Supporting information

**S1 Consort Checklist. CONSORT 2010-Checklist and CONSERVE-CONSORT Checklist.**
(DOCX)

**S1 Protocol. WILL trial protocol.**
(PDF)

**S1 Appendix. WILL trial statistical analysis plan.**
(PDF)

**S2 Appendix. WILL health economics analysis plan.**
(DOCX)

**S3 Appendix. Tables A-K.**
(DOCX)

## Acknowledgments

We wish to thank the women who participated in the trial, and the large number of site contributors who made possible this study. We extend our sincere thanks to our independent Trial Steering (Tim Draycott [Chair], Graeme MacLennan, Lucy MacKillop, Paul Mannix, Ruth Unstead-Joss) and Data Monitoring (Diana Elbourne (Chair), Henk Groen (Vice-Chair), Deborah Harrington, Edile Murdoch, Sarah Stock, and David Odd) Committees for generously contributing their time and guidance.

### Disclaimer

The views expressed are those of the author(s) and not necessarily those of NIHR, the NHS, or the Department of Health and Social Care, UK.

### Legend for supplementary appendices

AE (adverse event), APEC (Action on Pre-eclampsia Charity), ARM (artificial rupture of membranes), AST or ALT (aspartate aminotransferase or alanine aminotransferase), BAPM (British Association of Perinatal Medicine), BCTU (Birmingham Clinical Trials Unit), BMI (body mass index), BP (blood pressure), BW (birthweight), CACE (complier average causal effect), CC (critical care), CI (confidence interval), CiG (Co-investigators Group), CONSORT (Consolidated Standards of Reporting Trials), CRF (case report form), CRN (Clinical Research Network), DAPS (directly accessed pathology services), DAU (day assessment unit), dBP (diastolic blood pressure), DMC (Data Monitoring Committee), DIC (disseminated intravascular coagulation), ECG (electrocardiogram), EDD (estimated date of delivery), fullPIERS (full Pre-eclampsia Integrated Estimate of Risk Score), GA (gestational age), GCP (Good Clinical Practice), GCS (Glasgow Coma Scale), GDM (gestational diabetes), GP (general practitioner), HDP (hypertensive disorder of pregnancy), HEAP (Health Economics Analysis Plan), HIE (Hypoxic Ischaemic Encephalopathy), HIV (human immunodeficiency virus), HRG (health-care resource groups), hrs (hours), ICF (informed consent form), ICU (intensive care unit), IMAG (diagnostic imaging), INR (international normalised ratio), IQR (interquartile range [25th percentile, 75th percentile]), IRR (incidence rate ratio), ISF (investigator site file), ISRCTN (International Standard Randomised Controlled Trials Number), ISSHP (International Society for the Study of Hypertension in Pregnancy), ITT (intention-to-treat), ITU (intensive care unit), MAP (mean arterial pressure), MAU (medical assessment unit), MBRRACE-UK (Mothers and Babies: Reducing Risk through Audits and Confidential Enquiries across the UK), NA (not applicable), NHS (National Health Service), NICE (National Institute of Clinical Excellence), NNT (number-needed-to-treat), PI (principal investigator), PIS (participant information sheet), PF (plain film), PPH (postpartum

haemorrhage), PPI (public-patient involvement), qSOFA (Quick Sequential Organ Failure), RCT (randomised controlled trial), RD (risk difference), REC (research ethics committee), REF (reference), RP (regular day or night admissions), RR (risk ratio), RSUSAE (Related Unezpected Serious Adverse Event), SAE (Serious Adverse Event), SAP (statistical analysis plan), sBP (systolic blood pressure), SD (standard deviation), SOP (standard operating procedure), SpO2 (peripheral arterial oxygen saturation), STM (senior trial manager), TIA (transient ischaemic attack), TMG (Trial Management Group), TSC (Trial Steering Committee), UoB (University of Birmingham), wks (weeks)

## Author Contributions

**Conceptualization:** Laura A. Magee, Jon Dorling, Marcus Green, Jennifer A. Hutcheon, Ben W. M. Mol, Joel Singer, Pollyanna Hardy, Clive Stubbs, James G. Thornton, Peter von Dadelszen.

**Data curation:** Katie Kirkham, Sue Tohill, Eleni Gkini, Catherine A. Moakes.

**Formal analysis:** Laura A. Magee, Katie Kirkham, Eleni Gkini, Catherine A. Moakes, Mishal Javed, Jesse Kigozi.

**Funding acquisition:** Laura A. Magee, Jon Dorling, Marcus Green, Jennifer A. Hutcheon, Ben W. M. Mol, Joel Singer, Pollyanna Hardy, Peter von Dadelszen.

**Investigation:** Laura A. Magee, Katie Kirkham, Sue Tohill, Catherine A. Moakes.

**Methodology:** Laura A. Magee, Katie Kirkham, Eleni Gkini, Catherine A. Moakes, Jennifer A. Hutcheon, Ben W. M. Mol, Joel Singer, Pollyanna Hardy, Clive Stubbs, James G. Thornton, Peter von Dadelszen.

**Project administration:** Laura A. Magee, Katie Kirkham, Sue Tohill, Eleni Gkini.

**Resources:** Laura A. Magee.

**Supervision:** Laura A. Magee, Katie Kirkham, Catherine A. Moakes, Jennifer A. Hutcheon, Clive Stubbs, Peter von Dadelszen.

**Validation:** Katie Kirkham, Eleni Gkini, Catherine A. Moakes.

**Visualization:** Laura A. Magee.

**Writing – original draft:** Laura A. Magee, Katie Kirkham.

**Writing – review & editing:** Sue Tohill, Eleni Gkini, Catherine A. Moakes, Jon Dorling, Marcus Green, Jennifer A. Hutcheon, Jesse Kigozi, Ben W. M. Mol, Joel Singer, Pollyanna Hardy, Clive Stubbs, James G. Thornton, Peter von Dadelszen.

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
