## [Editor Report · Decision Letter 0]

25 Apr 2024

Dear Dr Magee, 

Thank you for submitting your manuscript entitled "WILL (When to Induce Labour to Limit risk in pregnancy hypertension): a randomised trial to determine optimal timing of birth for women with chronic or gestational hypertension at term" for consideration by PLOS Medicine.

Your manuscript has now been evaluated by the PLOS Medicine editorial staff and I am writing to let you know that we would like to send your submission out for external peer review.

Please re-submit your manuscript within two working days, i.e. by Apr 29 2024.

Feel free to email me at lgaynor@plos.org or us at plosmedicine@plos.org if you have any queries relating to your submission.

Kind regards,

Louise Gaynor-Brook, MBBS PhD

Senior Editor

PLOS Medicine

---

## [Decision Letter · Decision Letter 1]

4 Jun 2024

Dear Prof. Magee,

Thank you very much for submitting your manuscript "WILL (When to Induce Labour to Limit risk in pregnancy hypertension): a randomised trial to determine optimal timing of birth for women with chronic or gestational hypertension at term" (PMEDICINE-D-24-01330R1) for consideration at PLOS Medicine. 

Your paper was evaluated by four independent reviewers, including a statistical reviewer, and was discussed among all the editors here and with an academic editor with relevant expertise. The reviews are appended at the bottom of this email and any accompanying reviewer attachments can be seen via the link below:

[LINK]

I’m pleased to invite you to revise the paper in response to the reviewers’ and editors' comments. Obviously we cannot make any decision about publication until we have seen the revised manuscript and your response, and we plan to seek re-review by one or more of the reviewers. 

We expect to receive your revised manuscript by Jun 25 2024 11:59PM. Please email me (lgaynor@plos.org) if you have any questions or concerns.

We look forward to receiving your revised manuscript. 

Sincerely,

Louise Gaynor-Brook, MBBS PhD

plosmedicine.org

lgaynor@plos.org

Comments from the editors:

One of the subject reviewers has commented in their review: “It is unclear to the non-statistician reader how the statement "The lower limit of the aRD 95% CI did not include the pre-specified effect size of an 8 percentage point reduction' translates to their final conclusion 'As such, despite our lower-than-planned sample size, we were not underpowered to find target differences in our co-primary outcomes.'” Please revise this statement to substantiate your conclusion relating to underpower, in a manner that will be well understood amongst a general readership. 

It is important that the upper and lower boundaries of the non-inferiority margins selected are made clear and justified in the manuscript. 

Comments from the reviewers:

Reviewer #1: REVIEW WILL study PLOS one

Criteria for publication

In addition to being conducted in accordance with ethical standards and scientifically valid, research published in PLOS Medicine should fulfill each of the following criteria:

1. The research question is an important one to the community of researchers in this general area. 

Yes, it does, it is the largest international trial so far in the field of (early) near term induction of labour in chronic or gestational hypertension in a large multicenter, open-labeled, randomized trial including an economic cost analysis. After a thorough and well described analysis of the date, the authors propose earlier timing of delivery in case of both chronic and gestational hypertension in pregnancy based on the results of the WILL trial. 

2. The results provide a substantial advance over existing knowledge, with clear implications for patient care, public policy, or clinical research agendas. 

Yes, it does, it shows that early induction of labour at 38 + 0-3/7 weeks in case of mild chronic and gestational hypertension is non-inferior to care as usual regarding short term maternal and neonatal outcomes while adding potential benefits by reducing the number of Caesarean sections. The research group is planning an intentional patient data analysis mainly to corroborate the latter, as lower then expected adverse event rates do not justify mounting another randomized trial. 

3. Published together with an Author Summary written for general readers, the article is of interest to clinicians and policymakers who are not specialists in this topic. 

Yes, it changes current treatment protocols regarding chronic or gestational hypertension in pregnancy by adding new information underpinning shared decision making between patient and doctors regarding optimal timing of delivery. Based on the results of this study it has become clear that this proposed change in protocol will potentially reduce number of caesarean sections without an increase in neonatal admissions to the NICU, as well as save costs and reduce outpatient visits in an already overloaded national health system.

Review

* What are the main claims of the paper and how significant are they for the discipline? 

The main claim of the WILL trial is that planned early term birth at 38+0-3 weeks may be the preferred clinical option to do in case of chronic or gestational hypertension in pregnancy. It is the first national and largest international RCT in this field in a diverse study population making the results generalisable to real-world care of women in the United Kingdom. The WILL trial shows that in women with chronic or gestational hypertension who reach term and remain well, planned early term birth at 38+0-3 weeks, vs. usual care at term, averagely results in a six days earlier delivery. However, the expectant management group still requires iatrogenic induction of labour in 78% before spontaneous onset of labour. The authors conclude that planned early term birth at 38+0-3 weeks is non-inferior regarding adverse maternal and fetal/newborn co-primary outcomes compared with usual care at term. Planned early term birth (vs. usual care at term) did not increase (and may potentially reduce) Caesarean. There was a reduction in pre-eclampsia (defined broadly) for the woman, with no evidence of increased health problems for the baby. Finally, healthcare utilisation (e.g., monitoring of maternal-fetal wellbeing and obstetric outpatient visits) and associated costs were lower in the planned early term birth (vs. usual care at term) group, while the direction of costs overall favoured planned early term birth.

* Are the claims properly placed in the context of the previous literature? 

Yes, comparisons between the current and previous relevant literature have been made appropriately and addressed at. The discussion has however focused on short term outcomes and in that context, the authors state that the 6 days earlier time of birth is not likely to cause harm. The authors refer to the Cochrane Data Base of Systematic Review on Induction of Labour at or beyond 37 weeks of gestation, which results are stated to underpin this assumption. A similar conclusion has (very recently) been made by Grobman based on the ARRIVE trial data, saying that " Even when the only indication for delivery is the achievement of a full-term gestational age, evidence suggests that multiple different outcomes, including cesarean delivery, hypertensive disorders of pregnancy, neonatal respiratory impairment, and perinatal mortality, are less likely when induction is performed. This information underscores the importance of making the preferences of pregnant individuals for different birth processes and outcomes central to the approach to delivery timing".

Middleton P, Shepherd E, Morris J, Crowther CA, Gomersall JC. Induction of labour at or beyond 37 weeks' gestation. Cochrane Database Syst Rev. 2020;7:CD004945. 

Grobman WA. The role of labor induction in modern obstetrics. American Journal of Obstetrics and Gynaecology. 2024; 230(3), S662-S66. Doi.org/10.1016/j.ajog.2022.03.019.

* Have the authors treated the literature fairly? 

Not fully, as the authors do not mention any potential harmful long term effects on maternal or neonatal outcome, while reference to current literature (The Cochrane Data Base of Systematic Review on Induction of Labour at or beyond 37 weeks' of gestation) clearly states that data on long term effects looking at neurodevelopment at childhood are still lacking. The decision to induce labour is always a balance of risks and consequences for child and mother on the short as well as on the long term. In a relatively uncomplicated pregnancy before 40 weeks such as could be interpreted as described in the WILL study, long term effects of induction of labour on neurocognitive development should at least be discussed in the light of the existing literature. 

Bentley JP, Roberts CL, Bowen JR, Martin AJ, Morris JM, Nassar N. Planned Birth Before 39 Weeks and Child Development: A Population-Based Study. Pediatrics. 2016 Dec;138(6):e20162002. doi: 10.1542/peds.2016-2002. 

Burger RJ, Mol BW, Ganzevoort W, et al. Offspring school performance at age 12 after induction of labor vs non-intervention at term: A linked cohort study. Acta Obstet Gynecol Scand. 2023; 102: 486-495. doi:10.1111/aogs.14520

Davies-Tuck M., Wallace EM, Homer CSE. Why ARRIVE should not thrive in Australia, Women and Birth. 2018, 31(5), 339-340, doi.org/10.1016/j.wombi.2018.08.168.

Grobman WA. The role of labor induction in modern obstetrics. American Journal of Obstetrics and Gynaecology. 2024; 230(3), S662-S66. Doi.org/10.1016/j.ajog.2022.03.019.

Odd D, Glover Williams A, Winter C, Draycott T. Associations between early term and late/post term infants and development of epilepsy: A cohort study. PLoS One. 2018 Dec 31;13(12):e0210181. doi: 10.1371/journal.pone.0210181. 

* Do the data and analyses fully support the claims? If not, what other evidence is required? 

The claims stated can be concluded from the current study, even though the required number of inclusions has not been achieved.

* PLOS Medicine encourages authors to publish detailed methods as supporting information online. Do any particular methods used in the manuscript warrant such publication? If a protocol is already provided, for example for a randomized controlled trial, are there any important deviations from it? 

Yes, there is a study protocol available and there have been important deviations from it as the study has been stopped prematurely by the sponsor before including the desired number per group. Also, the primary stated desired gestational age of delivery in the expected management group changed from term to 39 weeks of gestation in the final phase of randomisation (17%). (On 11/August/2022 (after randomisation of 348 women), the control group was changed from 'expectant care until at least 40+0 weeks' to 'usual care at term', to reflect practice change related to timed birth in other populations(13, 14), the COVID-19 pandemic(15), and draft national labour induction guidance suggesting timed birth at 39 weeks may be appropriate for women at increased risk of term complications)

If so, have the authors explained adequately why the deviations occurred? Yes, the deviations and its potential effects have been adequately explained in the article. 

* Is this paper outstanding in its discipline? If yes, what makes it outstanding? If not, why not? 

Yes, it is outstanding as it adds novel insights to the current knowledge on optimal timing of delivery in case of constitutional or gestational hypertension at near term in the context of the National Health System of the UK. 

* Does the study conform to any relevant guidelines such as CONSORT, MIAME, QUORUM, STROBE, and the Fort Lauderdale agreement? 

Yes, a Consort checklist table has been provided. 

* Are details of the methodology sufficient to allow the experiments to be reproduced? 

Yes, there is sufficient detail to allow reproduction. 

* Is any software created by the authors freely available? 

Not applicable

* Is the manuscript well organized and written clearly enough to be accessible to non-specialists? 

Yes, it is well written and easy to read, in such a manner that non-specialists can sufficiently follow the train of thoughts of the authors in conducting the trial, understand the suggested adjustments of the study protocol and the effect thereof on the robustness of the data.

Reviewer #2: Statistical review

This paper reports a RCT evaluating an intervention of inducing labour early in women with hypertension. Due to the pandemic the trial was stopped early, although a sample size of 400 was recruited, meaning this trial provides useful data. The paper reported the trial very well and I only had a few minor comments below.

1. Abstract 'key secondary outcome was Caesarean' - should this be 'proportion of births that were Caesarean'?

2. Abstract: I note that one secondary outcome is reported in the abstract although there were multiple secondary outcomes. Generally it's considered good reporting practice to report all secondary endpoints or none in the abstract. Here the Caesarean outcome is described as a key secondary one so maybe it's appropriate to report it. However given the abstract could have additional words, it might be useful to add something like 'No other secondary outcomes showed notable differences between arms'.

3. Page 6, line 125: could the random element be provided?

4. Page 7, line 159: presumably the alpha was two-sided, could this be added? Could the alpha level for the non-inferiority outcome be added?

5. Page 7, it was not clear to me whether (in the original plan) both co-primary outcomes would have to be significant to recommend the intervention, or just one. Presumably the former given there was no multiple testing correction planned?

6. Results: I was surprised that no p-values (at least for the co-primary endpoints) were given, with the trial having pre-specified hypotheses. Perhaps the authors feel it is not appropriate given the trial didn't meet its enrolment criteria. Personally I would recommend they are given for the blue rows in Table 2, but if the authors feel strongly it's not warranted (especially if the SAP stated it wouldn't be done) then I'll not insist.

James Wason

Reviewer #3: This i a non -masked randomized trial in women with non-severe GHTN or CHTN enrolled at 36-37 weeks and allocated to either planned delivery at 38 weeks or until at least 40 weeks' gestation. There are 2 co-primary outcomes; One is composite poor maternal and one neonatal morbidity. The planned sample size was 1080 participants: 540/ each arm. Unfortunately, the trial was stopped prematurely by the Funding agency after 403 (< 40%) subjects were randomized due to poor recruitment. The authors conclude that on balance, planned early term birth at 38 weeks may be the preferred clinical option for women with CHTN or GHTN.

There are major issues with the manuscript:

1. Women with GHTN have a different pathophysiology than those with CHTN. Therefore, it was inappropriate to lump them together unless they had adequate sample size for each group with a planned secondary analysis.

2. Given, the premature termination of the trial, the conclusions in the abstract and manuscript should be changed to state that it the optimal timing of delivery for either group of women remains unknown.

3. It was inappropriate to change timing of delivery for the usual care group in the middle of the trial. 

4. The number of women with GHTN in this trial was only 208(105 v.103), and for CHTN it was only 195 (96 v. 99). Therefore, it is difficult to draw any conclusions or recommendations from this trial, particularly since the HYPITAT trial had more women with GHTN than this trial, and recommendation is to do delivery at 37 weeks' gestation. 

5. Almost 80% of enrolled subjects were receiving antihypertensive medications at enrollment, yet the authors considered these women to have non-severe hypertension. Thus, the authors should state that their findings apply only to women on medications, and does not address non- severe patients without medications.

Reviewer #4: This is an admirable study which aimed to answer an important and highly relevant question in clinical practice - the timing of birth management of this group of women who do not have preeclampsia is much more unclear than for those with preeclampsia. It is disappointing for the trial team especially that the trial was stopped early by the funder, it must have been extremely difficult conducting a trial of this scale during the pandemic anyhow.

Nil concerns about general methods - outlined in great detail and all seem appropriate for the intended analyses - however conclusions drawn from the findings of the primary analysis (8% aRD met) in the context of significantly smaller sample size needs further evaluation and discussion. It is unclear to the non-statistician reader how the statement "The lower limit of the aRD 95% CI did not include the pre-specified effect size of an 8 percentage point reduction' translates to their final conclusion 'As such, despite our lower-than-planned sample size, we were not

267 underpowered to find target differences in our co-primary outcomes.'

The limitations of the trial are documented openly however, a post-hoc power calculation would be useful to ascertain the significance of the actual trial findings. The recruitment of only 37% of the target should be included in the abstract, this is a major limitation.

The Discussion is comprehensive and the trial findings are well-placed in the context of current literature. The increase in development of preeclampsia (0.74 (0.56 to 0.98) in the control group has not been pulled out at all but should be discussed - this seems like a major finding.

Please ensure that the paper adheres to the PLOS Data Availability Policy (see http://journals.plos.org/plosmedicine/s/data-availability), which requires that all data underlying the study's findings be provided in a repository or as Supporting Information. For data residing with a third party, authors are required to provide instructions with contact information (web or email address) for obtaining the data. Please note that a study author cannot be the contact person for the data. PLOS journals do not allow statements supported by "data not shown" or "unpublished results." For such statements, authors must provide supporting data or cite public sources that include it. 

We expect all researchers with submissions to PLOS in which author-generated code underpins the findings in the manuscript to make all author-generated code available without restrictions upon publication of the work. In cases where code is central to the manuscript, we may require the code to be made available as a condition of publication. Authors are responsible for ensuring that the code is reusable and well documented. Please make any custom code available, either as part of your data deposition or as a supplementary file. Please add a sentence to your data availability statement regarding any code used in the study; eg “The code used in the analysis is available from Github [URL] and archived in Zenodo [DOI link]” Please review our guidelines at https://journals.plos.org/plosmedicine/s/materials-software-and-code-sharing and ensure that your code is shared in a way that follows best practice and facilitates reproducibility and reuse. Because Github depositions can be readily changed or deleted, we encourage you to make a permanent DOI’d copy (e.g. in Zenodo) and provide the URL. 

We ask every co-author listed on the manuscript to fill in a contributing author statement, making sure to declare all competing interests. If any of the co-authors have not filled in the statement, we will remind them to do so when the paper is revised. If all statements are not completed in a timely fashion this could hold up the re-review process. If new competing interests are declared later in the revision process, this may also hold up the submission. Should there be a problem getting one of your co-authors to fill in a statement we will be in contact. Please do not add or remove authors without first discussing this with the handling editor. You can see our competing interests policy here: http://journals.plos.org/plosmedicine/s/competing-interests. 

Please upload any figures associated with your paper as individual TIF or EPS files with 300dpi resolution at resubmission; please read our figure guidelines for more information on our requirements: http://journals.plos.org/plosmedicine/s/figures. While revising your submission, please upload your figure files to the PACE digital diagnostic tool, https://pacev2.apexcovantage.com/. PACE helps ensure that figures meet PLOS requirements. To use PACE, you must first register as a user. Then, login and navigate to the UPLOAD tab, where you will find detailed instructions on how to use the tool. If you encounter any issues or have any questions when using PACE, please email us at PLOSMedicine@plos.org. 

Please provide the name(s) of the institutional review board(s) that provided ethical approval.

Abstract: Please structure your abstract using the PLOS Medicine headings (Background, Methods and Findings, Conclusions). Please combine the Methods and Findings sections into one section, “Methods and findings”. 

At this stage, we ask that you include a short, non-technical Author Summary of your research to make findings accessible to a wide audience that includes both scientists and non-scientists. The Author Summary should immediately follow the Abstract in your revised manuscript. This text is subject to editorial change and should be distinct from the scientific abstract. Ideally each sub-heading should contain 2-3 single sentence, concise bullet points containing the most salient points from your study. In the final bullet point of ‘What Do These Findings Mean?’, please include the main limitations of the study in non-technical language. Please see our author guidelines for more information: https://journals.plos.org/plosmedicine/s/revising-your-manuscript#loc-author-summary. 

Please express the main results with 95% CIs as well as p values. When reporting p values please report as p<0.001 and where higher as the exact p value p=0.002, for example. Throughout, suggest reporting statistical information as follows to improve clarity for the reader “22% (95% CI [13%,28%]; p</=)”. Please be sure to define all numerical values at first use. 

Please include page numbers and line numbers in the manuscript file. Use continuous line numbers (do not restart the numbering on each page). 

Please cite the reference numbers in square brackets. Citations should precede punctuation. 

FIGURES AND TABLES 

Please provide titles and legends for all figures and tables (including those in Supporting Information files). 

Please define all abbreviations used in each figure/table (including those in Supporting Information files). 

Please consider avoiding the use of red and green in order to make your figure more accessible to those with color blindness. 

SUPPLEMENTARY MATERIAL 

Please note that supplementary material will be posted as supplied by the authors. Therefore, please amend it according to the relevant comments. 

Please cite your Supporting Information as outlined here: https://journals.plos.org/plosmedicine/s/supporting-information

REFERENCES 

PLOS uses the numbered citation (citation-sequence) method and first six authors, et al. 

Please ensure that journal name abbreviations match those found in the National Center for Biotechnology Information (NCBI) databases (http://www.ncbi.nlm.nih.gov/nlmcatalog/journals), and are appropriately formatted and capitalised. 

Where website addresses are cited, please include the complete URL and specify the date of access (e.g. [accessed: 12/06/2023]). 

Please also see https://journals.plos.org/plosmedicine/s/submission-guidelines#loc-references for further details on reference formatting. 

PLOS Medicine requires that all trials be prospectively registered in one of registries recognized by WHO. Please ensure that study registration details are included in the Methods section. 

Please structure the Methods section using the following sub-headings: Study design and participants, Randomization and masking, Procedures, Outcomes, Statistical analysis. 

Please ensure that all prespecified outcomes (primary, secondary, and exploratory) are listed in the Methods/Outcomes section and indicate whether there are outcomes that are not presented in the current report. 

Please specify the dates (Month Day, Year) during which study enrollment and follow up occurred. 

Please include absolute numbers wherever you report percentages; eg, n/N (%) 

Please present the safety data for the study including numbers of specific events and whether or not adverse events are thought to be related to treatment. AEs should be reported in the abstract, per CONSORT and CONSORT-Harms. 

Please complete the CONSORT checklist (https://www.equator-network.org/reporting-guidelines/consort/) and ensure that all components of CONSORT are present in the manuscript, including how randomization was performed, allocation concealment, blinding of intervention, definition of lost to follow-up, power statement. When completing the checklist, please use section and paragraph numbers, rather than page numbers. 

Please report your abstract according to CONSORT for abstracts, following the PLOS Medicine abstract structure (Background, Methods and Findings, Conclusions) https://www.equator-network.org/reporting-guidelines/consort-abstracts/

If your trial had to undergo important modifications in response to extenuating circumstances, please complete the CONSERVE-CONSORT checklist and provide in your Supporting Information; (https://www.equator-network.org/reporting-guidelines/guidelines-for-reporting-trial-protocols-and-completed-trials-modified-due-to-the-covid-19-pandemic-and-other-extenuating-circumstances-the-conserve-2021-statement/). When completing the checklist, please use section and paragraph numbers, rather than page numbers. 

Please add the following statement, or similar, to the Methods: "This study is reported as per XXXX guideline (S1 Checklist)." 

In keeping with our commitment to Open Science, please include the original study protocol document (as approved by the ethics committee) and analysis plan (including any amendments) as Supporting Information to be published with the manuscript if accepted. 

Please note that PLOS Medicine requires prospective, public registration of a data sharing plan (as part of mandatory clinical trials registration) for all clinical trials that began enrollment on or after January 1, 2019, in accordance with ICMJE requirements. 

[LINK]

---

## [Decision Letter · Decision Letter 2]

8 Aug 2024

Dear Dr Magee,

Many thanks for submitting your manuscript "WILL (When to Induce Labour to Limit risk in pregnancy hypertension): a randomised trial to determine optimal timing of birth for women with chronic or gestational hypertension at term" (PMEDICINE-D-24-01330R2) to PLOS Medicine. The paper has been reviewed by subject experts and a statistician; their comments are included below and can also be accessed here: [LINK]

After discussing the paper with the editorial team and an academic editor with relevant expertise, I'm pleased to invite you to revise the paper in response to the reviewers' comments. We may send the revised paper to some or all of the original reviewers, and we cannot provide any guarantees at this stage regarding publication.

When you upload your revision, please include a point-by-point response that addresses all of the reviewer and editorial points in full, indicating the changes made in the manuscript and either an excerpt of the revised text or the location (eg: page and line number) where each change can be found. Please also be sure to check the general editorial comments at the end of this letter and include these in your point-by-point response. When you resubmit your paper, please include a clean version of the paper as the main article file and a version with changes tracked as a marked-up manuscript. It may also be helpful to check the guidelines for revised papers at http://journals.plos.org/plosmedicine/s/revising-your-manuscript for any that apply to your paper.

We ask that you submit your revision by Aug 29 2024 11:59PM. However, if this deadline is not feasible, please contact me by email, and we can discuss a suitable alternative.

Don't hesitate to contact me directly with any questions (lgaynor@plos.org). 

Best regards, 

Louise 

Louise Gaynor-Brook, MBBS PhD 

Senior Editor

PLOS Medicine

lgaynor@plos.org

Comments from the editors:

Please expand the discussion to include the points made by reviewer 3. Some of the comments appear to have been addressed in the rebuttal but not in the manuscript itself e.g. the change in the management of the control group. Please ensure that the reviewer's concerns are addressed fully in the discussion section.

Comments from the reviewers: 

Reviewer #1: The authors have added tekst in answer to the comment on long term outcomes 

They answer:

Thank you for raising this important caveat about long-term outcomes. To our Discussion/Interpretation, we have added the following statement: "Also, observational data have raised concerns that in the general population, across the whole range of gestational age at birth, gestational age has a strong, dose-dependent relationship with special educational needs, including mild learning disabilities such as dyslexia and attention deficit hyperactivity disorder[(32)]. While data from non-randomised comparisons of labour induction or expectant care at term have been reassuring with regards to neurodevelopmental outcomes[(33, 34)], condition-specific data are needed. WILL participants were asked for consent for collection of routinely-collected health data, including those measuring school performance" (page 36, lines 419-26). *

Furthermore, the authors have made thorough and sufficient comments and/or alterations in the article where deemed neccesary in reply to the feedback of the other reviewers. As such, the WILL study from professor Magee and co workers (PMEDICINE-D-24-01330_R2) is judged to be acceptable for publication in PLOS Medicine

Reviewer #2: Thank you to the authors for addressing my previous comments.

On my previous comment about secondary outcomes in the abstract, I apologise for not being precise in my request (I had meant to mention *all* secondary endpoints or none). 

I don't think it's appropriate to just have significant secondary endpoints reported in the abstract but acknowledge it's not easy to report them all propertly: the authors could address this by adding 'Other secondary endpoints did not show significant differences between groups'.

Reviewer #3: It is unfortunate that the authors did not respond to the suggested changes. I still do not believe that their conclusions are accurate given the several limitations of their study that I emphasized in my review. They have inadequate sample size , they lumped two conditions with different pathophysiology, they changed management of the control group , etc. 

The authors responses are not convincing and the literature they cite has no relevance to the above concerns.

Reviewer #4: All comments and concerns have been adequately addressed and appropriates additions made to the manuscript. Well done.

---

* Please upload any figures associated with your paper as individual TIF or EPS files with 300dpi resolution at resubmission; please read our figure guidelines for more information on our requirements: http://journals.plos.org/plosmedicine/s/figures. While revising your submission, please upload your figure files to the PACE digital diagnostic tool, https://pacev2.apexcovantage.com/. PACE helps ensure that figures meet PLOS requirements. To use PACE, you must first register as a user. Then, login and navigate to the UPLOAD tab, where you will find detailed instructions on how to use the tool. If you encounter any issues or have any questions when using PACE, please email us at PLOSMedicine@plos.org.

* If the data are not freely available, please describe briefly the ethical, legal, or contractual restriction that prevents you from sharing it in the Data Availability Statement. Please also include an appropriate contact (web or email address) for inquiries. Please note that a study author cannot be the contact person for the data.

* Please provide a copy of the original trial protocol (as provided to and approved by the ethics committee) and the original statistical analysis plan as supplementary files. Please confirm whether the SAP was finalised prior to data lock, and that data was not accessible to the investigators prior to finalisation of the SAP. Please also provide a copy of the ethical approval document as a supplementary file. 

FIGURES AND TABLES

SUPPLEMENTARY MATERIAL

REFERENCES

RCTs 

* PLOS Medicine requires that all trials be prospectively registered in one of registries recognized by WHO. Please ensure that study registration details are included in the Methods section.

* Please structure the Methods section using the following sub-headings: Study design and participants, Randomization and masking, Procedures, Outcomes, Statistical analysis.

* Please clarify and explain all discrepancies between the paper and protocol. If the outcomes were not prespecified in the protocol, please define them in the Methods (Outcomes section) as post hoc and explain why they were added. Post-hoc comparisons should be presented as hypothesis generating rather than conclusive.

* Please ensure that all prespecified outcomes (primary, secondary, and exploratory) are listed in the Methods/Outcomes section and indicate whether there are outcomes that are not presented in the current report.

* Please specify the dates (Month Day, Year) during which study enrollment and follow up occurred.

* Please include absolute numbers wherever you report percentages; eg, n/N (%)

* Please present the safety data for the study including numbers of specific events and whether or not adverse events are thought to be related to treatment. AEs should be reported in the abstract, per CONSORT and CONSORT-Harms.

* Please complete the CONSORT checklist (https://www.equator-network.org/reporting-guidelines/consort/) and ensure that all components of CONSORT are present in the manuscript, including how randomization was performed, allocation concealment, blinding of intervention, definition of lost to follow-up, power statement. When completing the checklist, please use section and paragraph numbers, rather than page numbers.

* Please report your abstract according to CONSORT for abstracts, following the PLOS Medicine abstract structure (Background, Methods and Findings, Conclusions) https://www.equator-network.org/reporting-guidelines/consort-abstracts/

* If your trial had to undergo important modifications in response to extenuating circumstances, please complete the CONSERVE-CONSORT checklist and provide in your Supporting Information; (https://www.equator-network.org/reporting-guidelines/guidelines-for-reporting-trial-protocols-and-completed-trials-modified-due-to-the-covid-19-pandemic-and-other-extenuating-circumstances-the-conserve-2021-statement/). When completing the checklist, please use section and paragraph numbers, rather than page numbers.

* In keeping with our commitment to Open Science, please include the study protocol document and analysis plan (including any amendments) as Supporting Information to be published with the manuscript if accepted.

* Please note that PLOS Medicine requires prospective, public registration of a data sharing plan (as part of mandatory clinical trials registration) for all clinical trials that began enrollment on or after January 1, 2019, in accordance with ICMJE requirements.

---

## [Decision Letter · Decision Letter 3]

23 Sep 2024

Dear Prof. Magee,

Thank you very much for re-submitting your manuscript "WILL (When to Induce Labour to Limit risk in pregnancy hypertension): a randomised trial to determine optimal timing of birth for women with chronic or gestational hypertension at term" (PMEDICINE-D-24-01330R3) for review by PLOS Medicine.

I have discussed the paper with my colleagues and the academic editor and it was also seen again by one reviewer. I am pleased to say that provided the remaining editorial and production issues are dealt with we are planning to accept the paper for publication in the journal.

[LINK]

We expect to receive your revised manuscript within 1 week. Please email me (lgaynor@plos.org) if you have any questions or concerns.

We look forward to receiving the revised manuscript by Sep 30 2024 11:59PM.   

Sincerely,

Louise Gaynor-Brook, MBBS PhD

Senior Editor 

PLOS Medicine

plosmedicine.org

Requests from Editors:

General comments:

Throughout the paper, please adapt reference call-outs to the following style: "... every year [1,2]." (noting the absence of spaces within the square brackets).

In trials, there is usually a distinction in the language in terms of causal vs associational for primary and secondary trial outcomes. It would be beneficial to use associational language for secondary outcomes.

Where results are presented as e.g. “27, 13% vs. 24, 12%”, please revise for clarity. 

To help us extend the reach of your research, please provide any Twitter handle(s) that would be appropriate to tag, including your own, your coauthors’, your institution, funder, or lab.

Data availability:

Within your Data Availability Statement, please describe briefly the ethical, legal, or contractual restriction (e.g., data contain potentially identifying or sensitive patient information) that prevents you from sharing it in a public repository or as Supporting Information at the time of article publication. It would be preferable to have a more general institutional email address provided as an independent point of contact, for example BCTUdatashare@contacts.bham.ac.uk (as shared within the data sharing outline request in your supplementary materials).

Regarding code sharing, PLOS encourage authors to provide a synthetic (dummy) dataset that allows code to be shared without breaching data restrictions. Please note that this is not currently a requirement for publication, and we leave this to the discretion of the authors. 

Title: Please revise your title according to PLOS Medicine's style. Your title must be nondeclarative and not a question. It should begin with the main concept if possible. We suggest “Determining optimal timing of birth for women with chronic or gestational hypertension at term: The WILL (When to Induce Labour to Limit risk in pregnancy hypertension) randomised trial” or similar

Abstract:

Abstract Background: Please define WILL at first use.

Abstract Methods and Findings:

It would be preferable for the Abstract to be written as prose. 

Please ensure that all numbers presented in the abstract are present and identical to numbers presented in the main manuscript text.

Please provide brief demographic details of the study population (e.g. age, ethnicity, relevant characteristics, etc)

Please define UK at first use (apologies that this is obvious).

Line 64 - please revise “(27, 13% vs. 24, 12%” for clarity [assumedly this is n=27 (13%) vs n=24 (12%)]

Line 66 - It would be beneficial to use associational language for secondary outcomes.

Line 69 - please revise “(14, 7% vs. 14, 7%, respectively” for clarity

Line 70 - It would be beneficial to use associational language for secondary outcomes.

Line 71 - please revise for clarity 

In the last sentence of the Abstract Methods and Findings section, please describe 2-3 of the main limitations of the study's methodology.

Please state that analysis was intention to treat.

Please include length of follow up. 

Please provide the number of participants lost to follow up in each group.

Abstract Conclusions:

Please begin your Abstract Conclusions with "In this study, we observed ..." or similar, to summarize the main findings from your study, without overstating your conclusions. Please emphasize what is new and address the implications of your study, being careful to avoid assertions of primacy. 

Please provide the trial registration in full (ISRCTN77258279).

Author Summary:

Line 88 - please revise to ‘seven in 100 women’

Line 102 - Please use associational language for secondary outcomes.

In the final bullet point of ‘What Do These Findings Mean?’, please describe the main limitations of the study in non-technical language.

Introduction (please revise ‘Background’)

Line 134 - please define WILL at first use within the main manuscript.

Methods:

Please include your Ethics statement in the Methods section.

Please add the following statement, or similar, to the Methods: "This study is reported as per the CONsolidated Standards Of Reporting Trials (CONSORT) guideline and Guidelines for Reporting Trial Protocols and Completed Trials Modified Due to the COVID-19 Pandemic and Other Extenuating Circumstances (CONSERVE-CONSORT) (S1 Checklist)." When completing the checklists, please use section and paragraph numbers, rather than page numbers which will no longer correspond to the appropriate sections after copy-editing.

Please define "lost to follow-up" as used in this study. Other reasons for exclusion should be defined.

The following outcomes measures appear to differ between the submitted manuscript and the original protocol: 

Aspirin use is not reported in the manuscript as a secondary outcome, as specified in the protocol 

A number of secondary outcomes have been reported in the manuscript but not pre-specified in the original protocol. We note that these were specified in the SAP prior to data lock. These include Maternal sepsis; home visits; A&E visits; respiratory morbidity; Clinical respiratory problem, defined as meconium aspiration syndrome, pneumonia, pneumothorax/pneumomediastinum, transient tachypnoea of newborn, or other [unspecified]); Chest x-ray, N performed, N abnormal and nature of abnormality (i.e., meconium aspiration syndrome, pneumonia, pneumothorax/pneumomediastinum, transient tachypnoea of newborn, or other [unspecified]); HIE, defined as therapeutic hypothermia for ≥72 hours; Sepsis requiring antibiotics for at least five days, with confirmed blood or cerebrospinal fluid culture; Major operation (laparotomy, thoracotomy, craniotomy, or other); Birthweight; Neonatal apgar scores

Please specify in the manuscript when the additional outcomes were added (relative to trial completion and data lock) and why they were added (e.g. was a related study published that informed this, etc). Please also indicate in the manuscript whether the additional outcomes were approved by the IRB/ethics committee. 

Can results relating to aspirin use be presented as part of this manuscript? If not, please indicate why that is not possible. 

Please present the safety data for the study including numbers of specific events and whether or not adverse events are thought to be related to the intervention.

Results: 

Line 267 - GA at initiation of birth and GA at birth both appear to differ by 0.9 weeks according to Table 2, but the manuscript text does not reflect this. Please clarify. 

Line 268 - please provide provide 95% CIs and p value

Lines 298, 309 - please revise results presented for clarity.

Lines 315-329 - Please use associational language for secondary outcomes; please revise results presented for clarity.

Lines 350-357 - Please use associational language for secondary outcomes.

Please define the length of follow up (eg, in mean, SD, and range).

Please provide the actual numbers of events for the outcomes, not just summary statistics or ORs.

When a p value is given, please specify the statistical test used to determine it (in the accompanying table legend will suffice).

Discussion:

Please present and organize the Discussion as follows: a short, clear summary of the article's findings; what the study adds to existing research and where and why the results may differ from previous research; strengths and limitations of the study; implications and next steps for research, clinical practice, and/or public policy; one-paragraph conclusion.

Please remove all subheadings within your Discussion e.g. Summary of findings

Lines 372-379 - Please use associational language for secondary outcomes.

Lines 381, 443 - please temper assertions of primacy by adding ‘to the best of our knowledge’ or similar.

Lines 414-420 - sentence is very long; please consider revising by separating into two sentences.

Line 443-450 - sentence is very long; please consider revising by separating into two sentences.

Tables (including the supplementary tables):

Table 2 - please indicate in the legend what is represented by the numbers for GA at initiation of birth / at birth 

Please provide the unadjusted comparisons as well as the adjusted comparisons in Tables 2 and 3.

When a p value is given, please specify the statistical test used to determine it in the table legend.

Tables S8, S9 - please be clear that the risk ratios provided are adjusted risk ratios. Please provide the unadjusted comparisons. 

References:

Please ensure that journal name abbreviations match those found in the National Center for Biotechnology Information (NCBI) databases (http://www.ncbi.nlm.nih.gov/nlmcatalog/journals), and are appropriately formatted and capitalised.

Supplementary files: 

You may use almost any description as the item name of your supporting information as long as it contains an "S" and number. For example, “S1 Appendix” and “S2 Appendix,” “S1 Table” and “S2 Table,” and so forth. 

Please use whole numbers when naming your supporting information files. Combine separate parts (e.g., S1A and S1B Table) into one file (e.g. S1 Table) or rename with whole numbers (e.g., S1 and S2 Table).

Comments from Reviewers:

Reviewer #3: No comments.

[LINK]

---

## [Editor Report · Decision Letter 4]

30 Sep 2024

Dear Dr Magee, 

On behalf of my colleagues and the Academic Editor, Prof. Gordon Smith, I am pleased to inform you that we have agreed to publish your manuscript "Determining optimal timing of birth for women with chronic or gestational hypertension at term: The WILL (When to Induce Labour to Limit risk in pregnancy hypertension) randomised trial" (PMEDICINE-D-24-01330R4) in PLOS Medicine.

PRESS

Sincerely, 

Louise Gaynor-Brook, MBBS PhD 

lgaynor@plos.org